# Gearbox Fault Diagnosis Based on Optimized Stacked Denoising Auto Encoder and Kernel Extreme Learning Machine

Zhenghao Wu [1], Hao Yan [1], Xianbiao Zhan [1,2], Liang Wen [1] and Xisheng Jia [1,*]

[1]  Shijiazhuang Campus, Army Engineering University of PLA, Shijiazhuang 050003, China;
sttwhao@163.com (Z.W.); yhfy_1997@163.com (H.Y.); zxblzx999@163.com (X.Z.); lwenmark@163.com (L.W.)
[2]  Hebei Key Laboratory of Condition Monitoring and Assessment of Mechanical Equipment,
Shijiazhuang 050003, China
*   Correspondence: Correspondence: xs_jia2022@163.com

**Abstract:** The gearbox is one of the key components of many large mechanical transmission devices. Due to the complex working environment, the vibration signal stability of the gear box is poor, the fault feature extraction is difficult, and the fault diagnosis accuracy makes it difficult to meet the expected requirements. To solve this problem, this paper proposes a gearbox fault diagnosis method based on an optimized stacked denoising auto encoder (SDAE) and kernel extreme learning machine (KELM). Firstly, the particle swarm optimization algorithm in adaptive weight (SAPSO) was adopted to optimize the SDAE network structure, and the number of hidden layer nodes, learning rate, noise addition ratio and iteration times were adaptively obtained to make SDAE obtain the best network structure. Then, the best SDAE network structure was used to extract the deep feature information of weak faults in the original signal. Finally, the extracted fault features are fed into KELM for fault classification. Experimental results show that the classification accuracy of the proposed method can reach 97.2% under the condition of low signal-to-noise ratio, which shows the effectiveness and robustness of the proposed method compared with other diagnostic methods.

**Keywords:** stacked denoising automatic encoder; kernel extreme learning machine; gearbox; fault diagnosis

## 1. Introduction

The gearbox is a key component of the transmission system of self-propelled artillery, tanks, helicopters, and other equipment, and plays an important role in the operation of equipment. These pieces of equipment operate in complex and harsh environments such as long-term wind and sand, rain and snow, and plateau hypoxia, which lead to gear wear, broken teeth and other faults. If these faults are not detected and handled as soon as possible, the operation of the equipment may be affected, resulting in a reduction in the working efficiency of the transmission system, or the equipment may be damaged, resulting in casualties and other serious accidents. The vibration signal contains all the information of the equipment's running state. Fault prognostics and health Management (PHM) technology based on vibration signal can better improve the reliability and safety of equipment and avoid serious faults of equipment. Therefore, the use of vibration signal for gearbox fault diagnosis has important practical value [1–3].

Health monitoring and fault diagnosis of equipment by extracting features from vibration signals is a common method used by many scholars [4–6]. The purpose of extracting features from vibration signals is to use them as indicators to judge whether a gearbox has failed or deteriorated in function. For example, Li et al. [7] used Pearson correlation coefficient plots and orthogonality to select time domain features of diesel engines and used the parameters of the best better-correlated time domain features to input into a generalized regression neural network for fault classification, which could significantly improve the accuracy of fault identification. Meng et al. [8] used time domain

and frequency domain features to input into the KELM optimized via quantum particle swarm optimization (QPSO) for fault classification, which could well identify gearbox faults. Zhao et al. [9] used adaptive variational time domain decomposition (VTDD) to decompose and reconstruct complex multi-shock vibration signals. This method was used to extract features from the vibration signals of engine valve faults with good results. Wang et al. [10] used Pearson correlation coefficients and fault state selection features to be able to diagnose engine faults and damage levels well. Dhamande et al. [11] used continuous and discrete wavelets to extract time–frequency domain features from gear and bearing vibration signals for fault diagnosis, with better results than conventional time domain and frequency domain feature parameters. Li et al. [12] generated depth statistical features from the time, frequency, and time–frequency domain signals of rotating machinery and fed them into a support vector machine (SVM) for fault diagnosis with good results.

The above methods using feature extraction can solve some fault diagnosis problems well; however, they are often not effective in dealing with complex vibration signals. Complex vibration signals are often mixed with a lot of noise and have non-smooth, non-linear characteristics. In order to solve this problem, the method of signal decomposition was used by many scholars. Examples include empirical modal decomposition (EMD), variational modal decomposition (VMD), local mean decomposition (LMD), wavelet transform (WT), etc. Yang et al. [13] combined a hybrid fault diagnosis method of EMD and WT, which can effectively solve the problem of component failure in wind turbine gearboxes. Yan et al. [14] can eliminate the noise in the original signal well using VMD. Han et al. [15] used LMD to decompose the original signal into a finite number of product functions (PF) and extracted multi-scale symbolic dynamic information entropy (MSDE) as features for fault diagnosis; the method was experimentally proven to be good at eliminating noise. Syed et al. [16] used the eight decomposition coefficients of the wavelet family to extract the root-mean-square energy as a feature, which was fed into the classifier with very satisfactory results. These methods can well eliminate the noise in the vibration signal and retain the main fault characteristic signals, effectively improving the fault diagnosis accuracy of complex vibration signals.

Currently, deep learning-based fault diagnosis methods are a hot research topic. Deep learning differs from traditional methods in denoising signals. In some deep learning models, vibration signals are converted into time–frequency images and features in time–frequency images are extracted directly, which can avoid complex noise reduction and redundant feature elimination. For example, Liu et al. [17] used CNN to extract the features of continuous wavelet transform (CWT) to con-struct the wavelet time–frequency graph, which was input into the ELM classifier optimized using the Sparrow search optimization algorithm (SSA), which could well identify bearing faults. Gao et al. [18] used convolutional neural networks to extract features of time–frequency images, so as to recognize various fault modes of compressors. Some deep learning uses encoders for noise reduction. For example, Du et al. [19] used the convolutional autoencoder network to capture information from the eigenmatrix encoded by the natural gas consumption sequence with good results. Jia et al. [20] used the sparrow search algorithm (SSA) to optimize the SDAE network and input the time–frequency map generated by the CWT into the optimized SDAE for fault diagnosis, with a high accuracy rate.

These methods, although solving the problem of gearbox fault diagnosis to a certain extent, also suffer from the following problems:

1. Traditional fault diagnosis methods need manual feature extraction. However, manual feature extraction is obviously time-consuming and laborious, and feature selection depends on past experience, which has limitations in practical engineering applications.
2. Deep learning methods can effectively learn the deep information hidden in the data, but the selection of parameters for commonly used deep learning network models is based on previous experience and personal experimental debugging, which is time-consuming and labor-intensive, and fault diagnosis models require a large amount

of labeled data to be trained for a long time to ensure the accuracy of the diagnosis results and the generalization of the diagnosis model.

A diagnostic model with fast diagnostic speed, high diagnostic accuracy, and strong operability is required for practical engineering applications. The commonly used method of choosing network parameters is based on human experience. However, this method is time-consuming and laborious, and the parameter selection cannot reach the optimal effect. Therefore, we introduce a network hyperparameter adaptive selection strategy based on optimization algorithm. Firstly, the PSO algorithm is improved to provide faster and better global optimization capabilities. The fast determination of optimal parameters for SDAE network models was achieved using improved PSO. Then, the optimized SDAE network was used to extract the deep features of the original signals. Finally, the fast classification ability of KELM was used to classify the extracted deep features quickly. The major contributions of this paper include the following:

1. By improving the inertia weights of particles and adopting the PSO with adaptive weights, the parameter optimization of the SDAE network is faster and more effective.
2. Using the optimized SDAE network structure, deep-level features can be extracted directly from the original signal, avoiding the disadvantages of manual feature extraction.
3. The SAPSO-SDAE-KELM diagnostic model proposed in this paper solves the problems of noise reduction and dimensional catastrophe of the original signal, avoids the phenomenon of overfitting, and achieves rapid diagnosis of gearbox faults.

The rest of the paper is organized as follows: Section 2 introduces the theory related to the gearbox fault diagnosis method proposed in this paper. Section 3 introduces the SDAE network construction and fault diagnosis process. Section 4 shows the experiments and data set generation. Section 5 shows the comparison of diagnostic methods and noise addition experiments, which validate the method proposed in this paper. Section 6 presents the conclusions of this paper and proposes future research plans.

## 2. Theoretical Background

### 2.1. SDAE Implements the Principle of Dimensionality Reduction and Denoising

SDAE is a deep-learning network composed of multiple denoising autoencoders. SDAE is an unsupervised learning method that can extract deep features of gearbox faults from vibration signals.

#### 2.1.1. Principle of Noise Reduction with Denoising Autoencoder

The denoising autoencoder (DAE) [21] is an improvement on the auto encoder (AE) [22]. Noise reduction is achieved by adding noise to the original vibration signal at the input, corrupting the original signal via noise, and then reconstructing a better input without noise using a sample containing noise [23,24]. The structure of the DAE is shown in Figure 1.

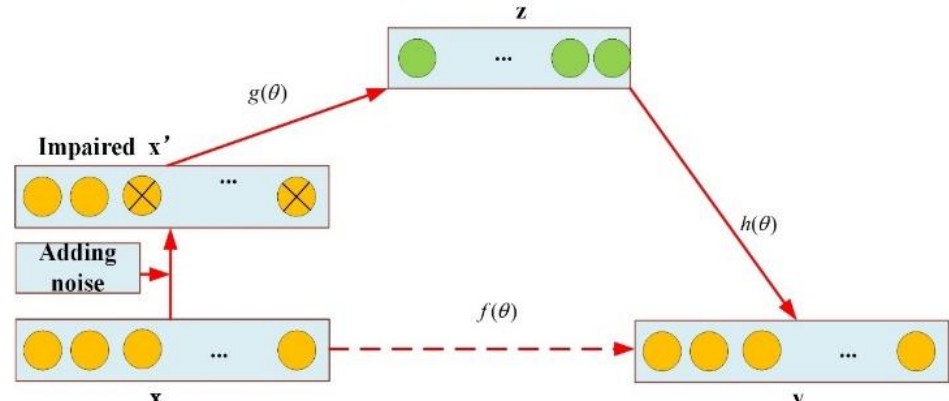

**Figure 1.** Structure of DAE.

The process of noise reduction of the original vibration signal by DAE is as follows:

1. Input noise to the original vibration signal $x$ to obtain the damaged signal $x'$.
2. The damaged signal a is used as the input, and the hidden layer Z is obtained through encoding. The encoding formula is as follows:

$$z = g(wx' + b) \tag{1}$$

3. After decoding and reconstruction, the reconstructed signal $y$ is obtained, so that reconstructed signal $y$ is close to the original signal $x$.

$$y = h(w\prime z + b') \tag{2}$$

4. Train the parameter $\theta = \{w, b, w', b'\}$ in the DAE with minimized reconstruction error:

$$L_{DAE} = \sum_{i=1}^{n} (x_i - y_i)^2 \tag{3}$$

where $x$ is the original vibration signal, $x'$ is the original signal after adding noise, $z$ is the hidden layer variable, $y$ is the reconstructed signal, $w$ is the weight coefficient, $b$ is the bias vector, $f(\bullet)$, $g(\bullet)$ and $h(\bullet)$ are the activation functions.

As can be seen from the above process, DAE noise reduction is achieved by adding noise to the original vibration signal, which, together with the noise of the original signal, is eliminated at the time of encoding. The hidden layer decoding provides a good restoration of the original vibration signal, and by minimizing the reconstruction error, the trained DAE model has better robustness.

2.1.2. Stacked Denoising Autoencoder (SDAE)

Although DAEs can achieve the effect of dimensional and noise reduction on the original vibration data, individual DAEs are still shallow neural networks that do not learn the deeper features of the original vibration signal very well. SDAE is a deep neural network made up of multiple DAEs stacked on top of each other. The hidden layer of the previous DAE is used as an input to the next DAE, forming a multi-layer DAE model structure, and finally achieving the extraction of feature information layer by layer; thus, SDAE has better learning ability and expression ability for data features. The structure of the SADE is shown in Figure 2.

The SDAE stacking process is as follows:

The first DAE training.

$$\begin{cases} z_1 = f(x') = s_f(W_1 x + b_1) \\ y = g(z_1) = s_g(W_2 z_1 + b_2) \end{cases} \tag{4}$$

The error reconstruction is calculated as follows:

$$L_{SDAE}(x, y) = \sum_{i=1}^{n} (x_i - y_i)^2 \tag{5}$$

The weighting and bias update formulae are as follows:

$$\begin{cases} W'_1 = W_1 - \tau \frac{\partial L_{SDAE}}{\partial y} \times \frac{\partial y}{\partial W_1} \\ b_1' = b_1 - \tau \frac{\partial L_{SDAE}}{\partial z_1} \times \frac{\partial z_1}{\partial b_1} \end{cases} \tag{6}$$

In the above equation, $\tau$ is the learning rate, $\frac{\partial L_{SDAE}}{\partial W_1}$ is the partial derivative of $W_1$, and $\frac{\partial L_{SDAE}}{\partial b_1}$ is the partial derivative of $b_1$. After the first DAE is trained, the hidden layer is extracted and used as the input layer for the next DAE.

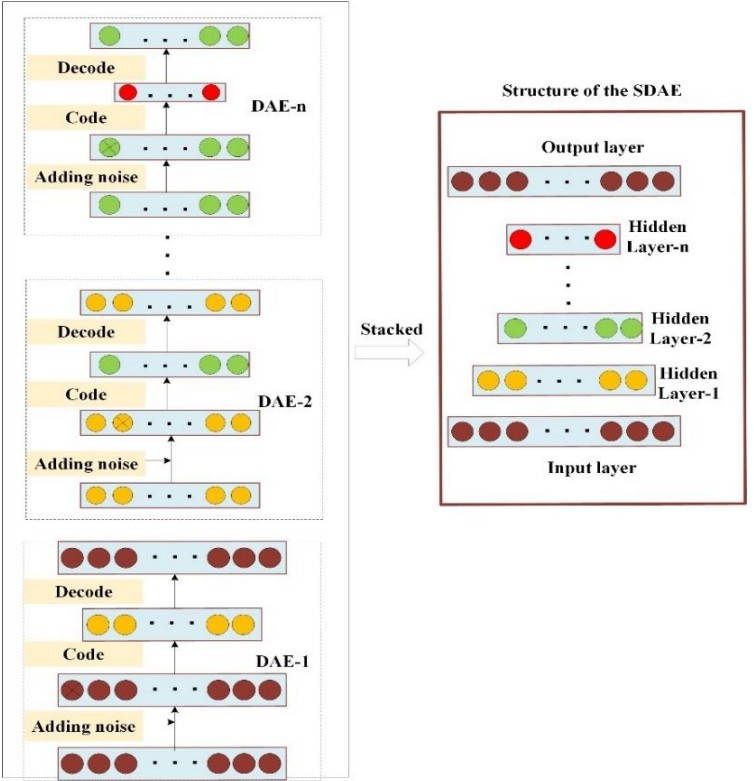

**Figure 2.** Structure of SDAE.

The second DAE training

$$\begin{cases} h_1 = f(x') = s_f(W_2 x' + b_2) \\ y = g(z_1) = s_g(W_3 z_2 + b_3) \end{cases} \tag{7}$$

The weights and bias updates are calculated as follows:

$$\begin{cases} W_2' = W_2 - \tau \frac{\partial L_{SDAE}}{\partial y} \times \frac{\partial y}{\partial W_2} \\ b_2' = b_2 - \tau \frac{\partial L_{SDAE}}{\partial z_2} \times \frac{\partial z_2}{\partial b_2} \end{cases} \tag{8}$$

The subsequent training process is the same as in steps 1 and 2, and all SDAEs in SDAE are trained in the same way. The weights and biases of the SDAE network are adjusted using a gradient descent algorithm [25] to fine-tune them in reverse.

### 2.2. An Improved PSO Algorithm for Selecting SDAE Network Parameters

The particle swarm optimization algorithm [26] is a swarm optimization method that simulates the predatory habits of birds. Before the optimization algorithm begins, a random initialization determines the initial velocity and position of the particles. Set the position of the $i$th particle in $n$-dimensional space to be $X_i = [x_{i,1}, x_{i,2} \ldots, x_{i,n}]$ and the initial velocity to be $V_i = [v_{i,1}, v_{i,2} \ldots, v_{i,n}]$; then, for each iteration of training, at moment t, the particle's velocity and position are expressed as in (12) and (13):

$$v_{i,j}(t+1) = \omega v_{i,j}(t) + c_1 r_1 [p_{i,j} - x_{i,j}(t)] + c_2 r_2 [p_{1,j} - x_{i,j}(t)] \tag{9}$$

$$x_{i,j}(t+1) = x_{i,j}(t) + v_{i,j}(t+1), j = 1, \ldots, n \tag{10}$$

where $c_1$ and $c_2$ are learning factors; $r_1$ and $r_2$ are uniformly distributed random numbers between 0 and 1, and $P_i = [p_{i,1}, p_{i,2} \ldots, p_{i,n}]$ is the best position in the local neighborhood.

The PSO algorithm is superior in solving complex optimization problems, but has the following drawbacks: the algorithm does not have high search accuracy, has a poor local search capability, tends to fall into local minimal value solutions, and has some dependence on parameters. In view of these drawbacks, the inertia weights $\omega$ of the particles are improved in this paper [27] by self-adopting the PSO algorithm with adaptive weights. The SAPSO algorithm balances global search capability with local improvement capability by taking non-inertial dynamic inertia weights, which are calculated as shown below:

$$\omega = \begin{cases} \omega_{\min} - \frac{(\omega_{\max} - \omega_{\min}) * (f - f_{\min})}{f_{avg} - f_{\min}}, f \leq f_{avg} \\ \omega_{\max}, f > f_{vg} \end{cases} \tag{11}$$

where $\omega_{\max}$, $\omega_{\min}$ denote the maximum and minimum values of $\omega$, respectively, $f$ denotes the current objective function value of the particle, $f_{avg}$ and $f_{\min}$ denote the current average and minimum objective values of all particles, respectively.

### 2.3. Kernel-Based Extreme Learning Machine (KELM)

The extreme learning machine (ELM) was proposed by Huang [28] and others based on the theory of Moore–Penrose generalized inverse as the main solution to the problems of the slow learning rate, long iteration time, and the need to set learning parameters such as learning step size and learning rate in advance in single-hidden layer feedforward neural networks (SLFNs). The ELM simply sets the appropriate number of nodes, generates all the parameters required for the corresponding implicit layer, and determines the weights of the output layer by means of a least squares method. ELM is widely used in engineering practice due to its advantages of fast learning and approximation of non-linear terms. Kernel-based extreme learning machine (KELM) [29] is an improved algorithm based on extreme learning machine and combined with kernel functions. KELM can improve the performance of the model while retaining the benefits of ELM. The basic structure of the extreme learning machine is shown in Figure 3.

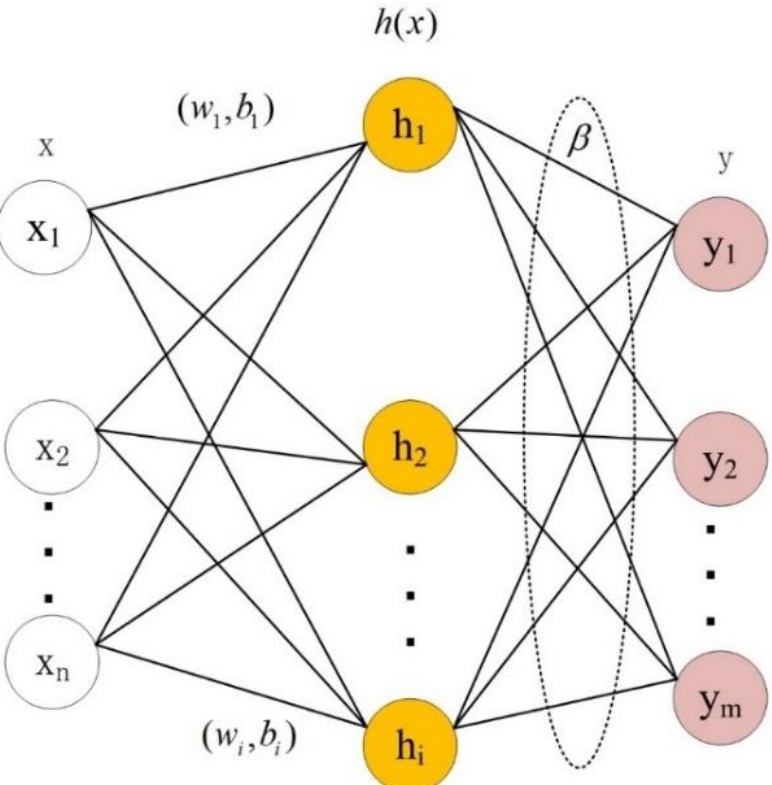

**Figure 3.** The basic structure of the ELM.

The model equation for the extreme learning machine is as follows:

$$y = h(x)\beta = H\beta \tag{12}$$

where $x$ is the input vector, y is the output vector, $h(x)$, H is the hidden layer feature matrix, and $\beta$ is the output weight.

$\beta$ is calculated as follows:

$$\beta = H^T(HH^T + \frac{I}{C})^{-1}T \tag{13}$$

where $T$ is the input vector, $C$ is the canonical factor and $I$ is the unit matrix.

KELM is the introduction of a kernel function to replace the mapping of hidden layer nodes in ELM and its model is calculated as follows:

$$y = h(x)H^T(HH^T + \frac{I}{C})^{-1}T = K(x)H^T(HH^T + \frac{I}{C})^{-1}T \tag{14}$$

Radial basis functions are selected as kernel functions in this paper:

$$\ker(x,y) = \exp(-\frac{\|x - y\|}{2\sigma^2}) \tag{15}$$

where $x$ is the input, $y$ is the output and $\sigma$ is the width parameter of the kernel function.

Introducing the kernel function into ELM results in the following output:

$$y = \begin{bmatrix} \ker(x,x_1) \\ \cdot \\ \cdot \\ \cdot \\ \ker(x,x_n) \end{bmatrix} H^T(HH^T + \frac{I}{C})^{-1}T \tag{16}$$

where $(x_1, x_2, \ldots, x_n)$ is the given training sample, which can be $n$ number of samples, and $\ker(\cdot)$ is the kernel function.

## 3. SDAE Network Construction and SAPSO-SDAE-KELM Troubleshooting Process

### 3.1. Construct the Optimal SDAE Network Chat Structure

#### 3.1.1. Determine the Number of Hidden Layers

Is it better to have more hidden layers in a SDAE neural network? For this problem, Du et al. [30] conducted simulation experiments and concluded that the network structure of SDAE containing three hidden layers for fault diagnosis works best. Experiments were carried out according to their experimental methodology, using data from the experiments in Section 4. The first thing that should be considered when determining whether SDAE hidden layer is optimal is the principle of SDAE. The essence of SDAE is to minimize the error value between the input data and the reconstructed data by decoding and encoding. The smaller the error between the input condition signal and the reconstructed data signal, the stronger the SDAE fault signal extraction capability, and the larger the error, the weaker the SDAE fault signal feature extraction capability. The smaller error value indicates that the sparse deep-level feature data in the hidden layer can well characterize the original data when the data in the hidden layer is used for fault diagnosis, which better reflects the rigor of the diagnosis method.

The RMSE characterizes the error between the input data and the reconstructed data in a very specific way; therefore, the root mean square error (RMSE) is a good evaluation metric to assess the magnitude of the error between the two vectors. Pearson's correlation coefficient is the most commonly used statistical measure to describe the extent to which two variables are related to each other. When the inputs and outputs are more similar, the error between the inputs and outputs is smaller; thus, the Pearson correlation coefficient

can be used as an evaluation indicator to measure the variation in inputs and outputs. Therefore, RMSE and Pearson's correlation coefficient are introduced in this paper as evaluation indicators. The calculation formula is as follows:

$$RMSE = \sqrt{\frac{1}{n}\sum_{i=1}^{n}(X_i - Y_i)^2} \tag{17}$$

$$\rho_{XY} = \frac{1}{n}\frac{\sum_{i=1}^{n}(X_i - E(X))(Y_i - E(Y))}{\sqrt{\frac{\sum_{i=1}^{n}(X_i - E(X))^2}{n}}\sqrt{\frac{\sum_{i=1}^{n}(Y_i - E(Y))^2}{n}}} \tag{18}$$

where $n$ is the number of vibration signal samples, $X_i$ is the original signal and $Y_i$ is the corresponding reconstructed signal. $E(X)$ is the mean value of the raw signal and $E(Y)$ is the mean value of the output signal.

In order to obtain the best-hidden layers for SDAE, SDAE networks containing 2, 3, 4, and 5 hidden layers are designed and verified, respectively. The RMSE, Pearson's correlation coefficient, and fault diagnosis accuracy were calculated, respectively. The details are shown in Table 1.

**Table 1.** Evaluation indicator values for different numbers of hidden layers.

| Number of Hidden Layers | 2 | 3 | 4 | 5 |
|:---|:---:|:---:|:---:|:---:|
| Pearson's Coefficient | 0.983 | 0.996 | 0.976 | 0.978 |
| RMSE | 0.0125 | 0.0093 | 0.0392 | 0.0534 |
| Diagnostic Accuracy | 99.33% | 100.0% | 98.0% | 96.67% |

As can be seen from the above table, the SDAE containing three hidden layers has a value of 0.0093 for RMSE, a correlation coefficient of 0.996, and a fault diagnosis accuracy of 100.0%. SDAE works best when it contains three hidden layers compared to other numb the errors of hidden layers. Therefore, the SDAE network structure with three hidden layers is adopted in this paper.

3.1.2. The Best Parameters of SDAE Are Selected via Improved PSO Optimization

The process of selecting parameters for improved PSO optimization of SDAE network structure is shown in Figure 4, and the specific steps are described as follows:

Step 1. Set the particle swarm velocity interval and position range.

Step 2. Initialize the population size and a number of iterations of the particle swarm optimization algorithm.

Step 3. Set the bit initial position of the particles. Calculate the fitness, which is determined by the root mean square error (RMSE) magnitude between the actual and predicted values of the particle population.

Step 4. A particle swarm search algorithm with a self-adaptive inertia factor $\omega$ is used for iterative optimization. In one iteration, the individual best extreme *pbest* of a single particle and the global best *gbest* of a population of particles are recorded, while the position and velocity of the particles are updated.

Step 5. Determine whether the current number of iterations exceeds the iteration limit. If yes, the iteration is terminated and the globally optimal result *gbest* and the *pbest* of individual particles are output. If no, the next step is executed.

Step 6. The optimal parameters are obtained. When the iteration proceeds until the RMSE of the actual and the predicted values are reduced to a predetermined value, the search for the best stops. At this point, the relevant parameters corresponding to the best position of the particle swarm are the best parameters in the SDAE network. If the RMSE is greater than the predetermined value, the iterative search continues and returns to step 4.

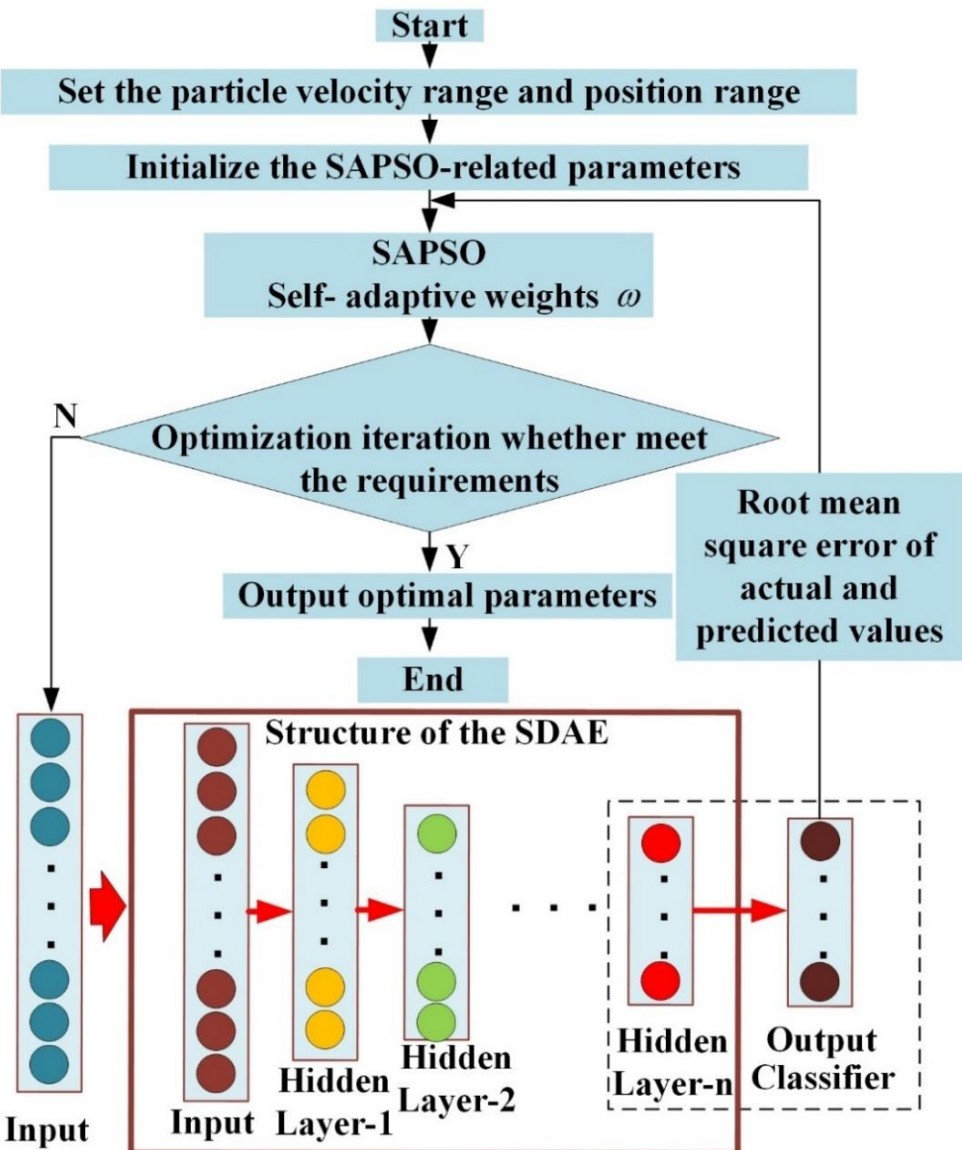

**Figure 4.** Flowchart of the improved particle swarm optimization algorithm for obtaining the optimal parameters of the SDAE network structure.

### 3.2. Fault Diagnosis Method and Process of SAPSO-SDAE-KELM

The SAPSO-SDAE-KELM-based gearbox fault diagnosis flow chart is shown in Figure 5 and is divided into the following three steps.

Step 1: Data pre-training. The vibration signal of the gearbox is obtained by using the experimental platform; the vibration signal is normalized, the redundant signal in the original signal is removed via PCA, and it is then divided into a training set and a test set.

Step 2: Unsupervised model training. The training set divided in step 1 is used to input the SAPSO-SDAE neural network model. The SDAE neural network model and its parameters are first initialized. The hyperparameter in the network model is then optimized via round-robin optimization. If the results of the network training do not meet the requirements, the training is re-iterated until the requirements are met.

Step 3: Supervised fault diagnosis. Using the parameters obtained in Step 2, the optimal SDAE network structure is constructed. The step 1 test set data are labeled and fed into the SDAE network for deep feature extraction. The data from the last hidden layer are extracted and fed into KELM for fault classification to produce fault diagnosis results.

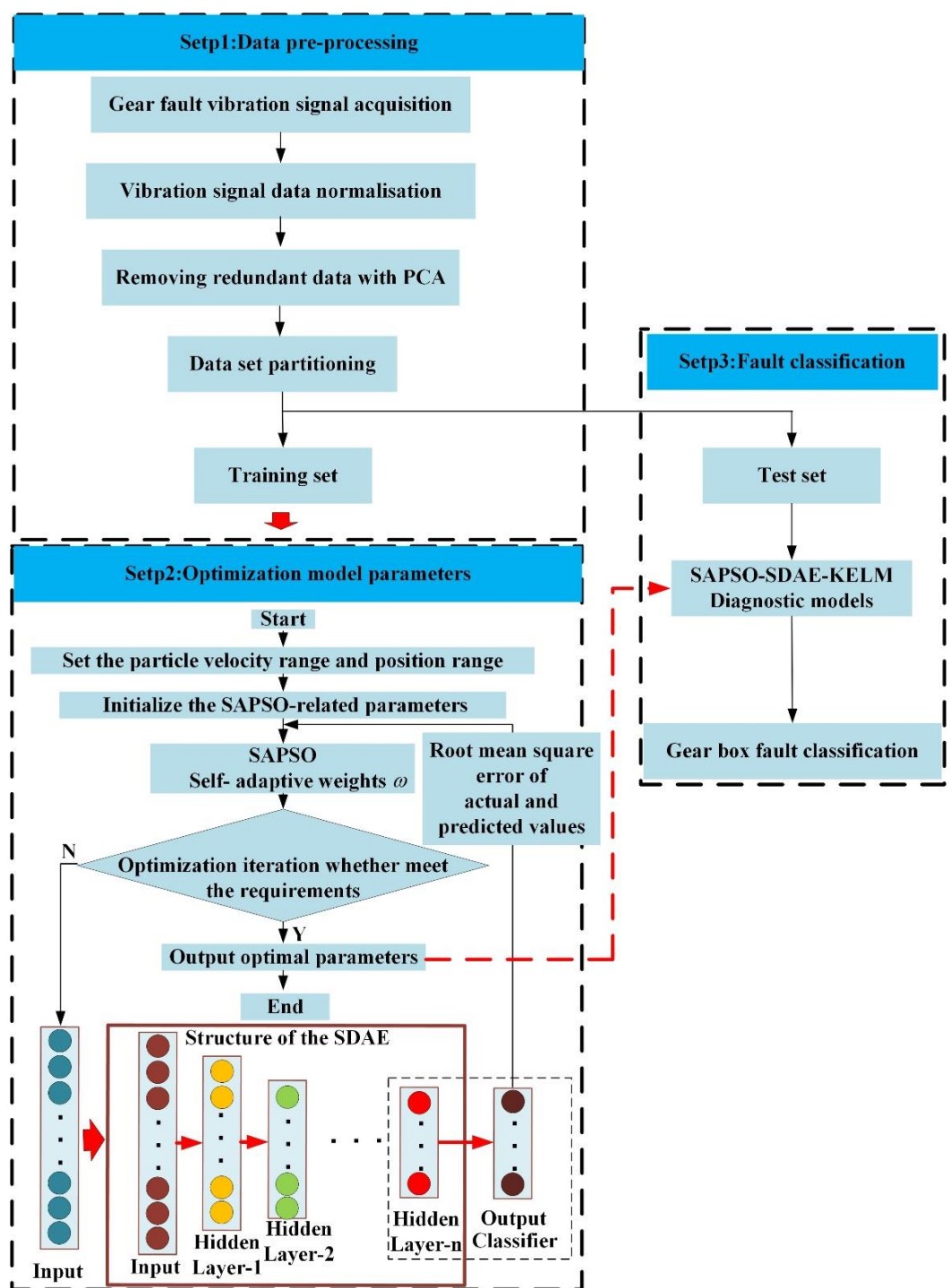

**Figure 5.** SAPSO-SDAE-KELM-based gearbox fault diagnosis flow chart.

## 4. Experiments and Data Pre-Processing

### 4.1. Experimental Platform

Figure 6 shows the structure and sensor layout of the test bed. The test bench consists of a base, planetary gear reducer, three-phase asynchronous motor, electromagnetic speed control motor, and magnetic powder brake. The experiments were carried out on a reducer model JZQ175. The vibration sensor model is IEPE general number. Using the CNC machine tool, the gearbox gear was preset as follows: 2 mm crack, 5 mm crack, and 2 mm broken teeth, and 5 mm broken teeth 4 kinds of faults. Figure 7 shows the preset fault.

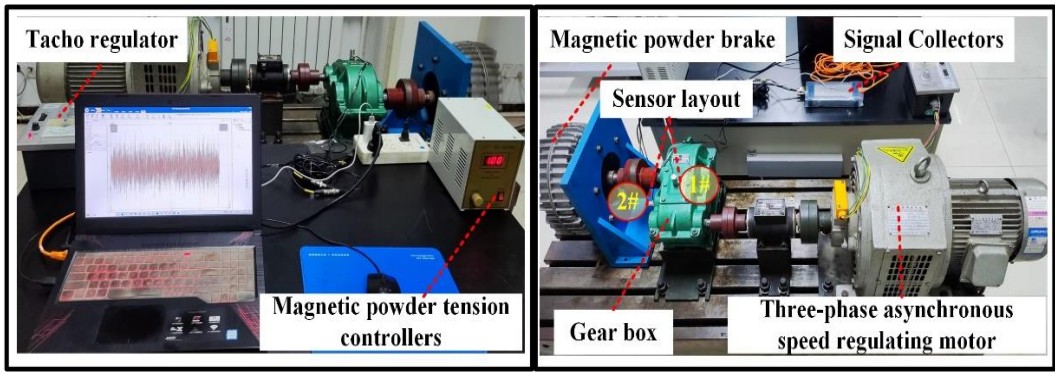

**Figure 6.** Test bench structure and sensor placement.

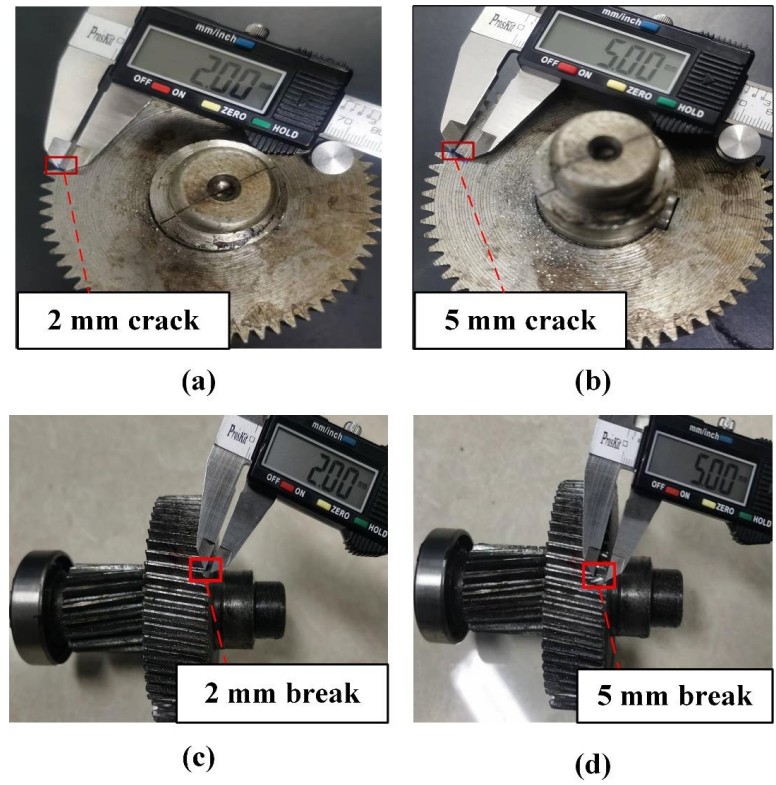

**Figure 7.** Fault preset condition: (**a**) 2 mm crack; (**b**) 5 mm crack; (**c**) 2 mm break. (**d**) 5 mm break.

*4.2. Signal Acquisition and Sample Generation*

4.2.1. Signal Acquisition Scheme

After the experimental platform runs smoothly, the signal was collected. The input speed of the motor was 1200 r/min, the magnetic powder brake load current was 1 A, the sampling frequency of the vibration sensor was 20,000 Hz, and the signal sampling time was 6 s. The vibration signals were collected for each of the five operating conditions.

4.2.2. Sample Construction and Data Set Generation

As can be seen from Table 2, 120,000 data points were collected for each operating conditions in this experiment. One sample was taken for every 800 points; therefore, 150 samples can be taken for each working condition, and the sample construction is shown in Figure 8.

**Table 2.** Vibration signal acquisition solution.

| Fault Status | Sampling Frequency | Sampling Time | Input Speed | Number of Sensors | Load Currents |
|---|---|---|---|---|---|
| normal | 20,000 Hz | 6 s | 1200 r/min | 2 | 1 A |
| 2 mm crack | 20,000 Hz | 6 s | 1200 r/min | 2 | 1 A |
| 5 mm crack | 20,000 Hz | 6 s | 1200 r/min | 2 | 1 A |
| 2 mm break | 20,000 Hz | 6 s | 1200 r/min | 2 | 1 A |
| 5 mm break | 20,000 Hz | 6 s | 1200 r/min | 2 | 1 A |

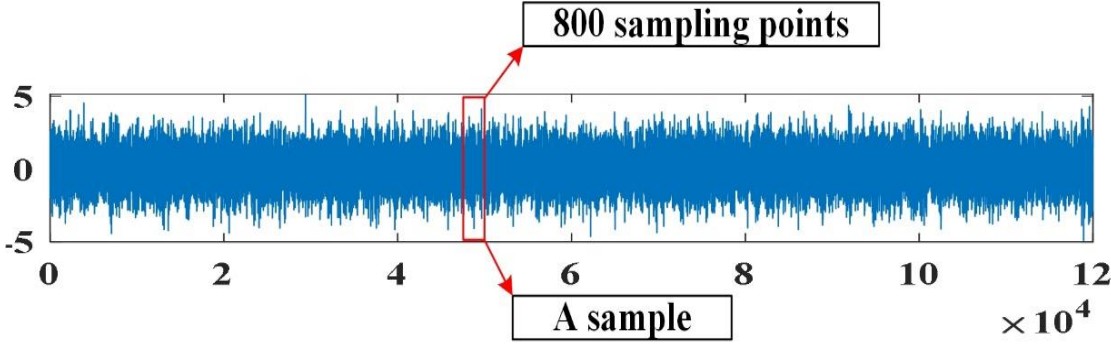

**Figure 8.** Sample construction.

In order to speed up the training, PCA is used to eliminate redundant signals from the original signal. Figure 9 shows the effect of PCA dimensionality reduction. It can be seen from Figure 9 that when the data are reduced to 112 dimensions, the essential characteristics of the original signal can be better retained. The ratio of training set and test set is divided into 8:2, and the details of sample division are shown in Table 3.

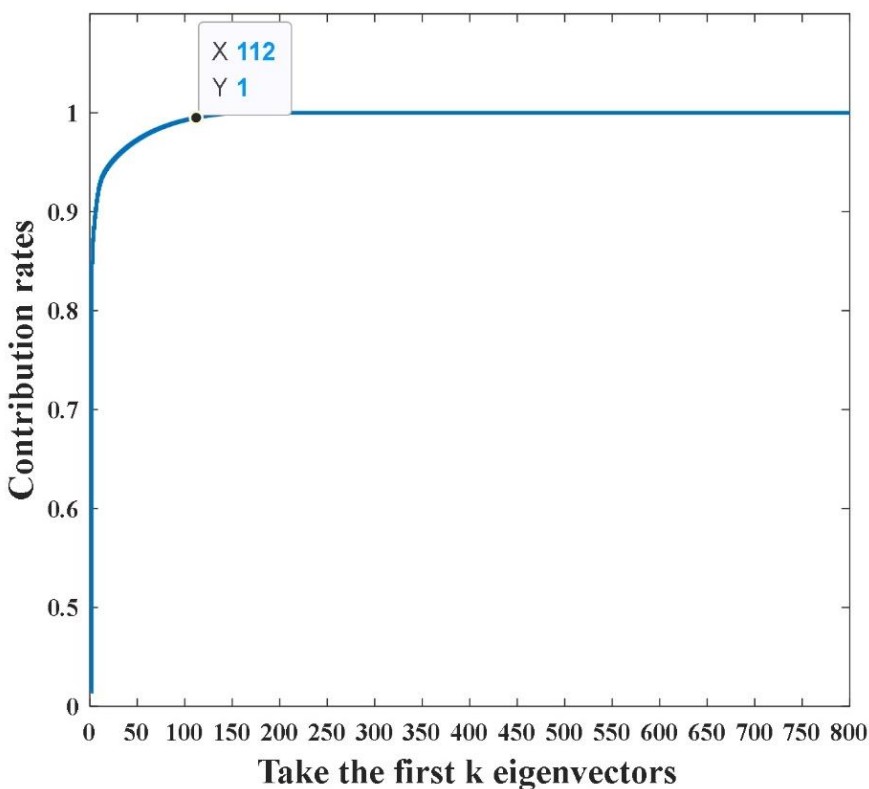

**Figure 9.** PCA reduces the dimension of the original signal.

**Table 3.** Composition of the dataset.

| Fault Status | Labels | Training Sets | Testing Sets |
| --- | --- | --- | --- |
| normal | L1 | $120 \times 112$ | $30 \times 112$ |
| 2 mm crack | L2 | $120 \times 112$ | $30 \times 112$ |
| 5 mm crack | L3 | $120 \times 112$ | $30 \times 112$ |
| 2 mm break | L4 | $120 \times 112$ | $30 \times 112$ |
| 5 mm break | L5 | $120 \times 112$ | $30 \times 112$ |

Training set and testing set according to the division of 8:2, as shown in Table 3.

## 5. Method Validation and Comparison

Before the method verification and comparison, the optimal SDAE neural network structure is first clarified. According to 3.1.1, the optimal network structure contains three hidden layers. Using the method in 3.1.2 and the sample data in 4.2, the optimal parameters of SDAE are obtained as follows. The number of iterations is 600, the learning rate is 0.6, the noise addition ratio is 0.3, and the number of hidden layer nodes is 69-55-46. The training curve of SDAE neural network is shown in Figure 10.

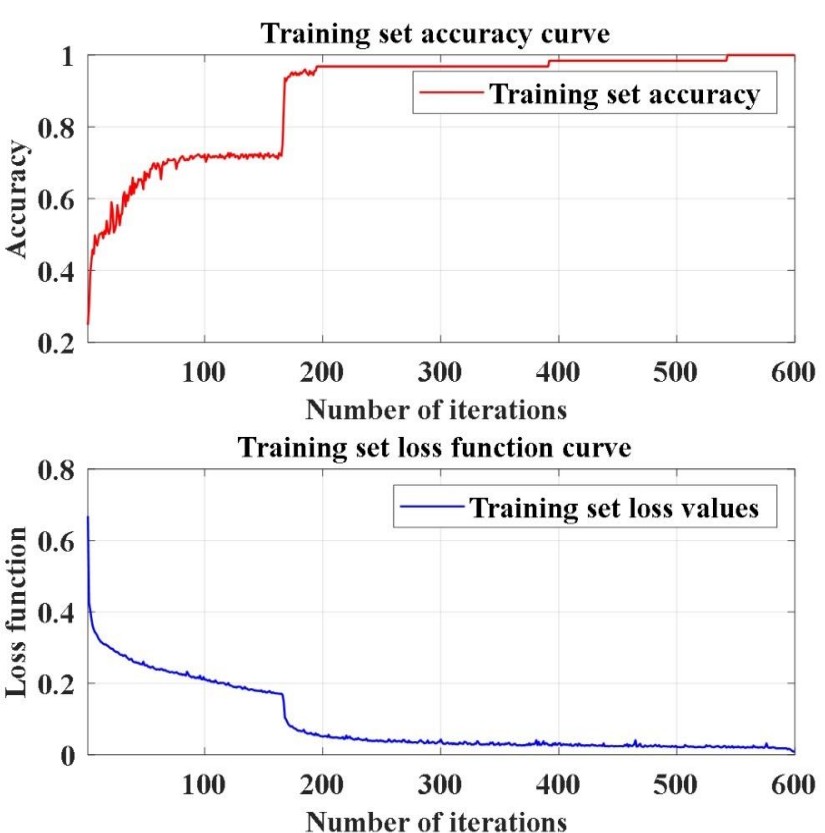

**Figure 10.** The SDAE training curve.

### 5.1. Comparison of Optimization Results after Particle Swarm Improvement

To verify that the improved PSO optimization method proposed in this paper is better, PSO and SAPSO are used to optimize the SDAE neural network, respectively. The RMSE of the actual value and the predicted value is used as the fitness value in 2.2; therefore, the size of the fitness value can be used to determine which optimization effect is better. The smaller the fitness value, the better the SDAE network is trained. Figure 11 shows the change of fitness value in the iterative optimization process of PSO and SAPSO. It can be seen from the figure that the final fitness value of PSO is 0.028, while the lowest fitness

value of SAPSO is 0.01. The fitness value of SAPSO is less than that of PSO, which can prove that SAPSO has a better optimization effect.

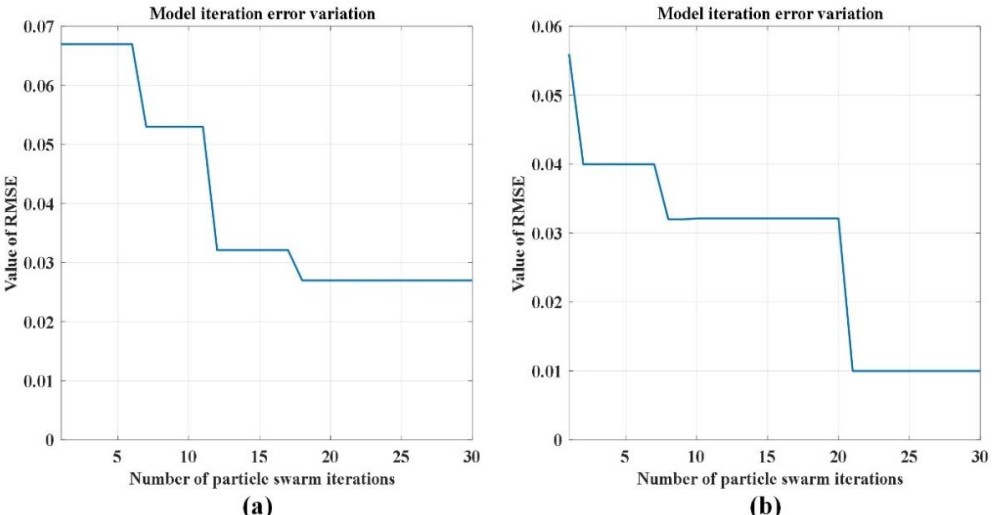

**Figure 11.** Variation of RMSE with number of iterations: (**a**) PSO; (**b**) SAPSO.

*5.2. Comparison with Other Fault Diagnosis Methods*

In order to verify the superiority of the SAPSO-SDAE-KELM diagnosis model proposed by the ontology, CNN-SSA-ELM, QPSO-KELM and PSO-SDAE-KELM fault diagnosis methods were selected to be compared with it. The vibration data collected via the above testbed were used for testing. These diagnostic methods include a deep learning method and a machine learning fault diagnosis method based on feature extraction. The diagnosis results are shown in Table 4. It can be seen from the table that the accuracy of the fault diagnosis method in this paper is 100% and the running time is 8.76 s. Compared with the other three methods, the fault diagnosis method in this paper has great advantages in diagnosis accuracy and diagnosis time.

**Table 4.** The accuracy and diagnosis time of different fault diagnosis methods.

| Labels | SAPSO-SDAE-KELM | PSO-SDAE-KELM | CNN-SSSA-ELM | QPSO-KELM |
|---|---|---|---|---|
| L1 | 100.0% | 100.0% | 100.0% | 100.0% |
| L2 | 100.0% | 100.0% | 100.0% | 100.0% |
| L3 | 100.0% | 100.0% | 90.0% | 90.0% |
| L4 | 100.0% | 93.33% | 83.33% | 76.67% |
| L5 | 100.0% | 93.33% | 90.33% | 93.33% |
| Diagnostic time | 8.71 s | 14.62 s | 10.33 s | 12.76 s |
| Accuracy | 100.0% | 97.33% | 93.33% | 92.0% |

It can be seen from Figure 12 that the accuracy of PSO-SDAE-KELM is slightly lower than that of SAPSO-SDAE-KELM, because SDAE feature extraction is greatly affected by parameters, while SAPSO is better than PSO in parameter optimization. The accuracy of CNN-SSA-ELM is 93.33%, and the diagnosis time is 14.36 s, which is the longest among these methods. Although CNN has a good effect in the feature extraction method, compared with SDAE, it lacks the effect of noise reduction, which affects the accuracy of fault diagnosis, and the long time image feature extraction requires leads to the slow speed of fault diagnosis. The fault diagnosis accuracy of QPSO-KELM is 92%, mainly because the fault features need to be extracted and selected manually in the diagnosis, which has a great impact on the speed and diagnosis results of fault diagnosis. From the above comparison, SAPSO-SDAE-KELM has superior performance in diagnostic accuracy.

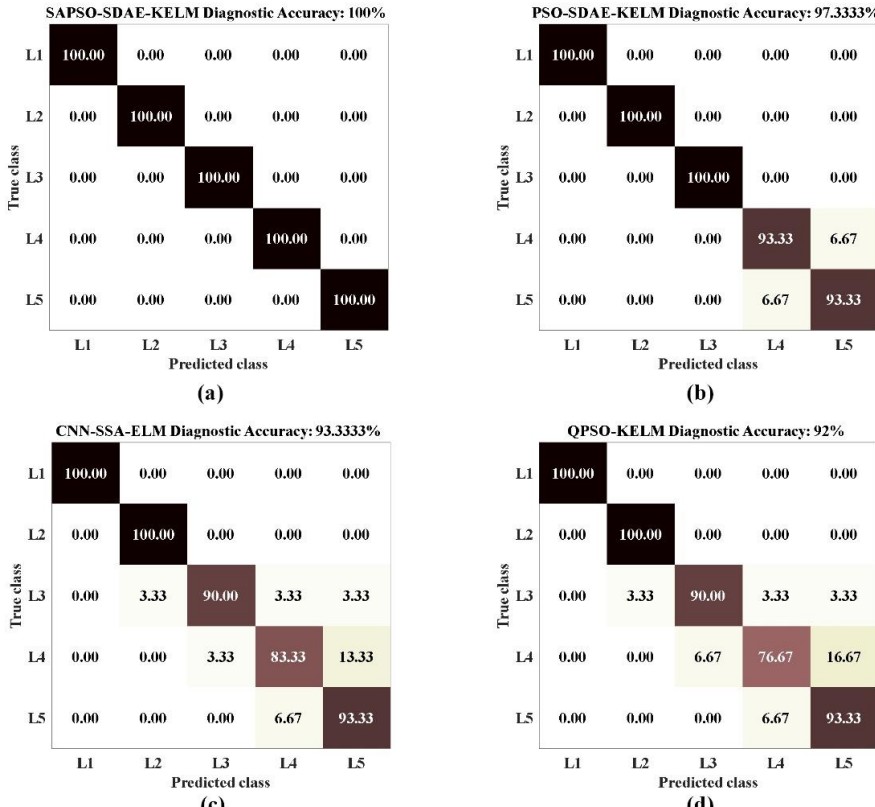

**Figure 12.** Accuracy of different diagnostic methods: (**a**) SAP-SO-SDAE-KELM; (**b**) PSO-SDAE-KELM; (**c**) CNN-SSSA-ELM; (**d**) QPSO-KELM.

### 5.3. Verification of Different Signal Inputs

To verify the generality of the proposed method in this paper, different signals are fed into the diagnostic model presented for validation. In this paper, the input signals are divided into original vibration signals, frequency domain signals and characteristic signals for testing.

The frequency domain signal is the original vibration signal obtained using the fast Fourier transform (FFT). The original vibration has 120,000 collection points for each working condition, and 60,000 points after transforming into frequency domain signal. Taking 400 points as a sample, there are 150 samples for each condition so that it is consistent with the original vibration signal data set. After a reduction to 78 dimensions using PCA, the training set is $120 \times 78$ and the test set is $30 \times 78$ for each condition.

The feature signal is used to extract the time domain features and frequency domain features of the original vibration signal as input signals. In this paper, 19 time domain features and 4 frequency domain features of the vibration signal are extracted as input signals, and the details are shown in Table 5.

**Table 5.** Vibration signal feature extraction situation.

| Feature Types | Extracted Features | Number of Features |
|---|---|---|
| Time domain feature | 1 maximum value, 2 minimum value, 3 peak–peak value, 4 mean value, 5 mean square value, 6 root mean square (RMS), 7 average amplitude, 8 root amplitude, 9 variance, 10 standard deviation, 11 peak value, 12 kurtosis, 13 skewness, 14 energy, 15 peak factor, 16 pulse factor, 17 waveform factor, 18 margin factor, 19 clearance factor. | 19 |
| Frequency domain feature | 1 frequency mean value, 2 frequency center, 3 root mean square frequency, 4 frequency standard deviation. | 4 |

Every 800 points in the original vibration signal are a sample for feature extraction so as to ensure that the number of samples in each working condition is 150, which is the same as the original vibration signal input samples. After 23 features were extracted, the input vector composed of each condition was $150 \times 23$, the training set was $120 \times 23$, and the test set was $30 \times 23$. After SAPSO optimization, the SDAE network parameters corresponding to the three signal input methods are shown in Table 6.

**Table 6.** SDAE parameters for different signal inputs.

| Input Signals | Number of Nodes in the Hidden Layer | Learning Rate | Noise Addition Rates | Number of Iterations |
|---|---|---|---|---|
| Time domain signals | 69-55-46 | 0.6 | 0.3 | 600 |
| Frequency domain signals | 56-42-34 | 0.4 | 0.1 | 300 |
| Feature signals | 15-11-6 | 0.5 | 0.1 | 200 |

In order to visualize how the SDAE network extracts deeper features from the data, the t- stochastic neighbor embedding (t-SNE) method is adopted to visualize the distribution of features in the last hidden layer of the SDAE neural network. The feature scatter plots of the hidden layers for the three different signal inputs are shown in Figure 13. As can be seen from the figure, the deep features of the five different operating conditions of the time domain signal extracted using SDAE can be well separated and there is no cross-mixing. The frequency domain signals and feature signals are fed into the SDAE network for deep feature extraction. From the graph, it can be seen that there is a small amount of crossover on the features labelled as L4 and L5; the other features can be well separated.

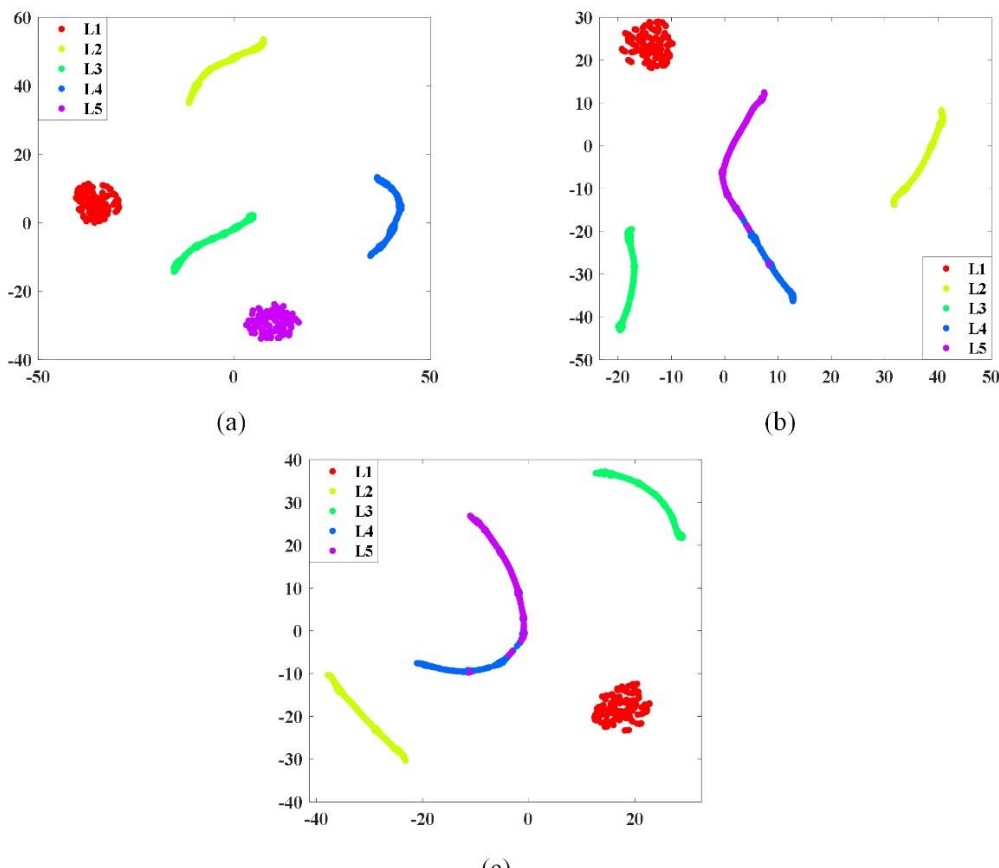

**Figure 13.** Feature visualization of hidden layers: (**a**) Time domain signals; (**b**) Frequency domain signals; (**c**) Feature signals.

Figure 14 shows the diagnostic results of the time domain signal, frequency domain signal and feature signal input into SAPSO-SDAE-KELM. The accuracy rates were 100%, 96.67% and 98.67%, respectively. The diagnostic results validate the results of the feature visualization in Figure 13 well. The lowest diagnostic accuracy of the three signal inputs is over 96%, indicating that the SDAE network can well extract the features of different signal inputs; therefore, the diagnostic model proposed in this paper has good generalization.

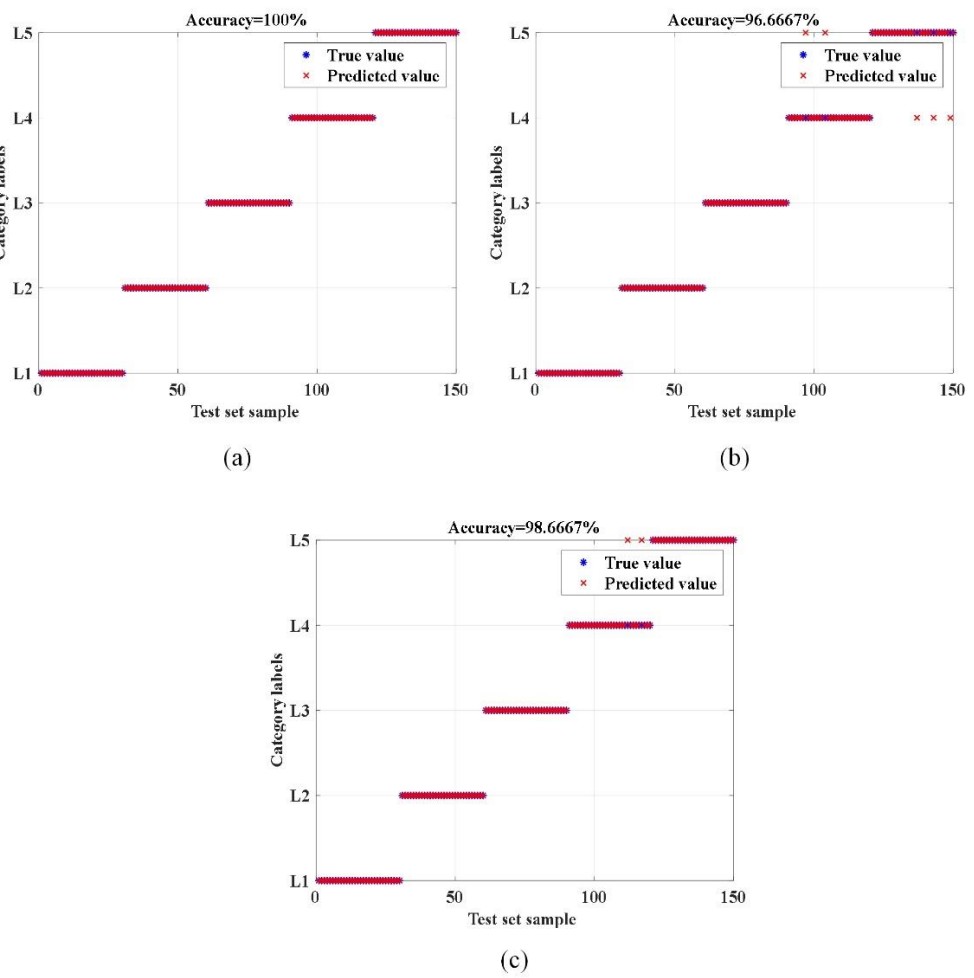

(a)

(b)

(c)

**Figure 14.** Diagnostic results for three different signals: (**a**) time domain signals; (**b**) frequency domain signals; (**c**) feature signals.

### 5.4. Verification of the Noise Reduction Effect

The actual working environment of gearboxes is very complex, facing the effects of different environments such as sand, rain, snow and plateau. The complex operating environment of the gearbox is simulated by adding noise to the original signal. By controlling the size of the input noise to test, add noise to the vibration signal to create the signal-to-noise ratios of −20 db, −15 db, −10 db, −5 db, 10 db in order to obtain different kinds of noise signals. Figure 15 shows the time domain waveform and spectrogram of the vibration signal after adding Gaussian white noise with different SNR in the case of 5 mm broken teeth. The figure shows that with the addition of noise, the amplitude of the vibration signal becomes significantly larger, thus showing that the fault signal is masked by the noise signal. Using signals containing different noises, they are fed into the fault diagnosis model proposed in this paper for fault diagnosis. In order to verify the stability of this model, the signals with added noise are fed into different diagnostic methods for comparison. The results are shown in Table 7.

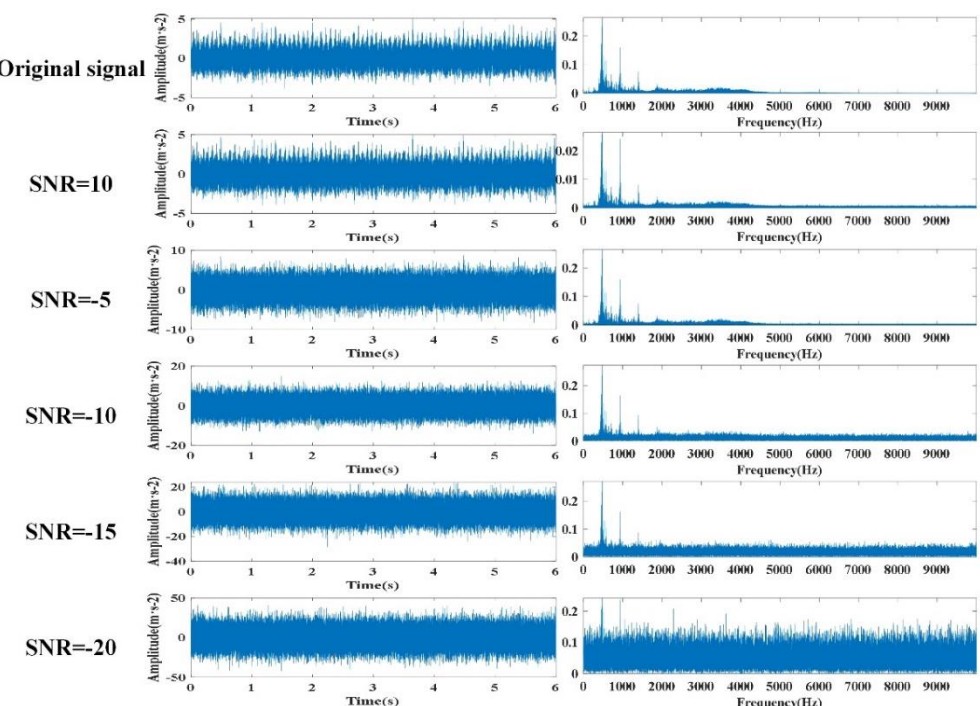

**Figure 15.** Time domain waveform and spectrum diagram under different signal-to-noise ratio.

**Table 7.** The diagnostic accuracy of different diagnostic methods under different signal-to-noise ratios.

| SNR (db) | Diagnostic Accuracy | | | |
|---|---|---|---|---|
| | SAPSO-SDAE-KELM | PSO-SDAE-KELM | CNN-SSSA-ELM | QPSO-KELM |
| 10 | 100.0% | 100.0% | 100.0% | 100.0% |
| −5 | 99.33% | 98.67% | 96.0% | 94.67% |
| −10 | 98.67% | 97.33% | 91.33% | 90.0% |
| −15 | 98.00% | 96.67% | 88.67% | 85.33% |
| −20 | 97.33% | 95.33% | 84.67% | 82.67% |

It can be seen from the table that the accuracy of the four diagnostic methods is affected by the noise, and the diagnostic accuracy decreases with the increase in the noise. The accuracy of PSO-SDAE-KELM is decreased by 4%, CNN-SSA-ELM is decreased by about 15%, QPSO-KELM is decreased by 17%, and SAPSO-SDAE-KELM is only decreased by 2.7%. The accuracy of the method proposed in this paper is still higher than 97% in the case of high noise, which shows that the model has a good anti-noise effect.

## 6. Conclusions

This paper presents a gearbox fault diagnosis model using SAPSO-SDAE-KELM. Through experimental comparison and analysis, the gearbox fault diagnosis model proposed in this paper outperforms other fault diagnosis models in terms of diagnostic accuracy and diagnostic speed. This paper's contributions are as follows:

1. The hyperparameters associated with the structure of the SDAE network have a significant effect on the classification effect of the model. The improved PSO was used to optimize SDAE and other parameters to realize the rapid adaptive adjustment of network structure.
2. The fault diagnosis is carried out by the optimized SDAE network with different signal inputs, and the diagnosis accuracy is above 96%, which proves that the diagnosis model in this paper has good generalizability corresponding to different signal inputs.

3. Through noise addition experiments, the method proposed in this paper has a high diagnostic accuracy in the presence of high noise. Compared to other diagnostic models, the method proposed in this paper has better noise immunity.

In this paper, we mainly study the use of SDAE to extract high-dimensional features of one-dimensional signals for fault diagnosis. The method in this paper can be used to distinguish different gearbox faults well, especially under the condition of low signal-to-noise ratio. The method also has high accuracy and good practicability. The experiments in this paper can only simulate the actual complex conditions to a large extent, and cannot represent the actual conditions. In the future, the gearbox hybrid fault diagnosis will be further studied.

**Author Contributions:** Conceptualization, Z.W., H.Y. and X.Z.; methodology, Z.W. and H.Y.; software, Z.W. and H.Y.; data curation, Z.W. and H.Y.; formal analysis, Z.W.; resources, Z.W., L.W. and X.J.; validation, Z.W. and X.Z.; visualization, Z.W.; investigation, Z.W., X.Z. and H.Y.; writing—originaldraft prepa-ration, Z.W.; writing—review and editing. Z.W., X.Z., L.W. and X.J.; funding acquisition, L.W. and X.J.; project administration, L.W. and X.J.; supervision, L.W. and X.J. All authors have read and agreed to the published version of the manuscript.

**Funding:** This research received no external funding.

**Data Availability Statement:** The data involved in this article have been presented in the article.

**Conflicts of Interest:** The authors declare no conflict of interest.

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
