# Peer review of "Gearbox Fault Diagnosis Based on Optimized Stacked Denoising Auto Encoder and Kernel Extreme Learning Machine"

_processes, doi:10.3390/pr11071936_

Round 1
Reviewer 1 Report
In this paper, a gearbox fault diagnosis method based on optimized stacked denoising auto encoder and kernel extreme learning machine is proposed. Then, the experimental data of gearbox is used for model verification. Here are some comments and suggestions:
1. In Abstract, the difficulty and significance of this work are not clear. The author should explain why the outside factors leading to increased difficulty. The quantitative results of this work should be also provided in the Abstract.
2. In Section 1, "Traditional machine learning methods need to extract features manually". Actually, the machine learning models like DNN, CNN or LSTM can extract features from the input data automatically, but the effect of feature extraction is different from each other. The author should revise this sentence to better description the significance of this work.
3. In Section 1, the authors introduce the traditional denoising methods and deep learning-based denoising method, however, the differences between these two methods are not clear.
4. For the deep learning-based autoencoder for feature extraction, some other scholars also conducted research. The author may also cite this reference in literature review: https://doi.org/10.1016/j.energy.2022.124689.
5. In Section 3, the figure 5 shows that the step two of the proposed method is unsupervised model training. However, according to the figure, the purpose of step is to obtain the optimal model parameters.
6. In figure 6, the RMSE between original signal and reconstructed signal is calculated. But the output of SDAE is actually the encoded results. The reconstructed results are acquired after encoding and decoding.
7. Some sentences have syntax error or expressions confusion. Please revised them:
l Therefore, the following work is done to improve this paper's time efficiency and fault diagnosis accuracy.
l the fault diagnosis model proposed in this paper can reduce the influence of noise in the original signal can better learn the deep-level features in the original signal and has higher diagnosis accuracy.
Minor editing of English language required
Author Response
请参阅附件

Reviewer 2 Report
I recommended this paper for publication pending minor revision. My comments are in below:
(1) The paper needs nomenclature to better understand the parameters in equations from (1) to (11) which reduces the number of pages.
(2) In the first page and line (45) of the introduction section, the word correlated is duplicated twice.
(3) The time domain of the waveform in Figure (15) has signal to noise ratio with positive and negative signs. What does it mean? Can you please use Fast Fourier Transform to see the original and fault of the signal diagnoses?
(4) Can the author put the publisher name of each reference in the references section?
Reviewer 3 Report
A very well written manuscript. Concise and clear.
Suggestions for improving the manuscript are as follows:
1. No Type of the Paper is selected. You should select Article,
2. All emails are from the 163.com domain. Why were institutional emails not entered?
3. Concrete results should be entered in the abstract.
4. Keywords are very long. It should be more explicit.
5. Grupno citiranje nije neophodno.
6. Estimate the measurement uncertainty of all measured results.
7. The measurements in Figure 7 are not clear. Display with a clearer figure. How reliable and accurate is this method of measurement?
8. Can a cost analysis be done?
9. State in the conclusions the possibilities of practical application.
10. State in the conclusions the limitations of the applied methodology.
11. State directions for future research in the conclusions.
Round 2
Reviewer 3 Report
The manuscript has been corrected